# Understanding the Biology and Immune Pathogenesis of Chikungunya Virus Infection for Diagnostic and Vaccine Development

**DOI:** 10.3390/v15010048

**Published:** 2022-12-23

**Authors:** Mohamad S. Hakim, Abu T. Aman

**Affiliations:** Department of Microbiology, Faculty of Medicine, Public Health, and Nursing, Universitas Gadjah Mada, Yogyakarta 55281, Indonesia

**Keywords:** chikungunya, diagnostic, immune responses, pathogenesis, vaccines

## Abstract

Chikungunya virus, the causative agent of chikungunya fever, is generally characterized by the sudden onset of symptoms, including fever, rash, myalgia, and headache. In some patients, acute chikungunya virus infection progresses to severe and chronic arthralgia that persists for years. Chikungunya infection is more commonly identified in tropical and subtropical regions. However, recent expansions and epidemics in the temperate regions have raised concerns about the future public health impact of chikungunya diseases. Several underlying factors have likely contributed to the recent re-emergence of chikungunya infection, including urbanization, human travel, viral adaptation to mosquito vectors, lack of effective control measures, and the spread of mosquito vectors to new regions. However, the true burden of chikungunya disease is most likely to be underestimated, particularly in developing countries, due to the lack of standard diagnostic assays and clinical manifestations overlapping with those of other endemic viral infections in the regions. Additionally, there have been no chikungunya vaccines available to prevent the infection. Thus, it is important to update our understanding of the immunopathogenesis of chikungunya infection, its clinical manifestations, the diagnosis, and the development of chikungunya vaccines.

## 1. Introduction

Chikungunya virus (CHIKV) is the responsible agent of chikungunya fever, a debilitating arthritic disease in humans. Acute infection of CHIKV is generally characterized by sudden onset of fever, rash, myalgia, and headache, which in some patients, progresses to severe and chronic arthralgia that persists for years [1]. Acute CHIKV infection symptomatically resembles the dengue virus (DENV) infection. Among differential diagnoses, a notable proportion of CHIKV disease is inaccurately diagnosed as dengue fever (DF) and other diagnoses, including leptospirosis, typhoid fever, enteritis, and non-specific viral infections [2].

CHIKV belongs to the *Alphavirus* genus within the *Togaviridae* family. *Alphavirus* causing inflammatory musculoskeletal diseases in humans is known as an “arthritogenic alphavirus”, which includes CHIKV, O’nyong-nyong virus (ONNV), Ross River virus (RRV), and among others [3]. CHIKV was initially isolated from an acute febrile patient during the Tanzanian outbreaks in 1952–1953 [4]. The viral genome consists of a positive-sense single-stranded RNA molecule measuring approximately 11.8 kb in length with a 5′-methylguanylate cap and a 3′-polyadenylate tail [5] (Figure 1). The CHIKV has spread globally and has been classified into four clades: East, Central, and South African (ECSA), West African (WA), Asian (AL), and Indian Ocean Lineages (IOL) [6,7].

The CHIKV disease is more frequently reported in tropical and subtropical regions. Recent outbreaks raise concerns about the future public health impact of CHIKV in temperate regions [8]. Urbanization, human travel, viral adaptation, lack of effective control measures, and spread of new vectors likely have contributed to the recent re-emergence of CHIKV infection [9]. In Indonesia, a recent study of children and adult patients presenting acute fever in eight hospitals confirmed CHIKV infection in 40 of 1089 screened subjects (3.7%) [2].

However, due to the lack of standard diagnostic assays, absence of typical signs and symptoms, and clinical manifestations overlapping with those of other infections, the true burden of CHIKV is most likely underestimated, particularly in developing countries [2]. CHIKV infection is normally a self-limiting disease with a low fatality rate (~0.1%). However, the frequent joint complications that lead to persistent disability have significant implications on public health in general, including a substantial impact on the quality of life (QOL) for infected patients as well as a burden in terms of economic and community perspectives [10]. Here, we discuss the recent understanding of the immune pathogenesis of CHIKV infection, its clinical manifestation, the diagnosis, and the development of CHIKV vaccines.

## 2. The Life Cycle of CHIKV

### 2.1. The Host Cell Receptors for CHIKV

Available evidence suggests that multiple pathways are employed for CHIKV entry in a cell-type specific manner [11]. One of the best characterized human receptors for CHIKV is the cell-adhesion molecule, matrix-remodeling-associated protein 8 (Mxra8), which is widely expressed in epithelial and mesenchymal cells [12,13,14]. Mxra8 is involved in the entry process of multiple arthritogenic alphaviruses, including CHIKV, ONNV, RRV, and Mayaro virus (MAYV) [15,16]. By creating various deletion variants, it has been shown that the stalk region of Mxra8 is crucial for facilitating CHIKV entry [12]. Another recent study showed that an insertion of 15 amino acids in the ectodomain of Mxra8 conferred resistance of *Bovinae* to CHIKV infection. Conversely, removal of this insertion resulted in enhanced susceptibility of *Bovinae* to CHIKV invasion [17]. In vitro, cell lines with low expression levels of Mxra8 were less susceptible to CHIKV infection [18]. In Mxra8-deficient mice, decreased CHIKV infection and CHIKV-induced joint swelling were observed [15]. In line with these findings, ectopic expression of Mxra8 in vitro in various cell lines and in vivo in *Drosophila* led to enhanced CHIKV invasion [15,16,18].

Studies elucidating the host cell receptors for CHIKV play an essential role in creating a path for the development of novel vaccines and entry inhibitors. Indeed, administration of Mxra8-Fc fusion protein and anti-Mxra8 monoclonal antibodies inhibited CHIKV infection both in vitro (in various cell lines) and in vivo (in mice) [16]. Similarly, mAb RRV-12 targeting the B domain of E2 glycoprotein blocked the infection of multiple alphaviruses, including CHIKV, MAYV, and RRV [19]. This domain is responsible for binding to Mxra8 [16,19]. Additionally, various in silico approaches have been employed to identify a potential inhibitor of Mxra8 [20,21]. Although many studies have shown that Mxra8 serves as a receptor for CHIKV, the absence or decreased expression of Mxra8 in several types does not completely block CHIKV infection. This finding suggests that CHIKV may employ other alternative cell receptors to invade the hosts.

Another host protein that has been demonstrated to be the cell receptor for CHIKV is CD147, also called basigin or extracellular matrix metalloproteinase inducer (EMMPRIN). CD147 is widely expressed in various human cell types, including fibroblast and endothelial cells. Similar to Mxra8, CD147 is involved in the replication process of multiple alphaviruses, including CHIKV, ONNV, RRV, MAYV, and Western equine encephalitis virus (WEEV), among others [22]. Remarkably, CD147 has a high structural homology with Mxra8, although further studies are needed to elucidate the precise molecular interactions between the CHIKV protein and CD147 [22].

Glycosaminoglycans (GAGs) were shown to be host molecules involved in the binding of CHIKV, particularly heparin/heparan sulfate (HS) [23,24]. Various CHIKV strains differ in their GAG utilization in the presence or absence of the Mxra8 receptor [23,25]. Interestingly, a single passage of CHIKV to mosquito (CHIKV_mos_) cell lines resulted in an attenuated phenotype, characterized by reduced replication and pathogenicity both in vitro and in vivo [26]. This phenotype was associated with a loss of binding to GAGs during passage in the mosquito cell lines and can be regained by passaging back in mammalian cells [26]. The B domain of the E2 glycoprotein was shown to be the receptor binding site and thus, responsible for binding to GAG-expressing cells [11]. Mutagenesis studies showed that the E2 residue 82 was a primary determinant of GAG utilization and influenced infectivity, viral dissemination, and immune-mediated joint pathology [25,27]. In addition to the E2 glycoprotein, polymorphisms in the E1 residue 156 and 211 glycoproteins also influenced binding to HS and modulated CHIKV-induced joint pathology [28]. Various other molecules, including prohibitin (PHB), T-cell immunoglobulin, and mucin domain 1 (TIM-1), have been demonstrated to be involved in the interactions with CHIKV [29,30,31].

### 2.2. The Entry Process of CHIKV

After binding to a receptor present on the cell surface, CHIKV enters the target cells by clathrin-mediated endocytosis [32]. This event is followed by membrane fusion, particularly via interaction with Rab5-positive endosomes. This fusion process is notably enhanced by the presence of cholesterol in the target membrane and is mediated by acidification of the pH in endosomes that eventually triggers penetration and uncoating of CHIKV [32]. Consistently, methyl β-cyclodextrin, a potent cholesterol depleting agent, was shown to significantly inhibit the infection of CHIKV. Similar results were obtained by treatment with lysomotropic agents inhibiting endosomal acidification, including chloroquine and bafilomycine [33]. Thus, clathrin-coated endocytic vesicles are able to penetrate the cell membrane and deliver the containing “cargo” of the CHIKV genome into the cytoplasm.

Clathrin-independent pathways have been reported to mediate CHIKV entry to the target cells [33,34]. Recently, macropinocytosis was demonstrated to be another entry pathway for CHIKV into human muscle cells [34,35]. Macropinocytosis is mediated by the formation of macropinosomes, which are large and uncoated vesicles involved in the unspecific uptake of extracellular material into the cytoplasm. Collectively, all of these studies indicate that CHIKV employs multiple host receptors and multiple host entry processes, enabling it to invade diverse cell types in multiple tissues.

### 2.3. CHIKV Replication within the Target Cells

A brief overview of CHIKV replication is depicted in Figure 2. Endosome acidification leads to membrane fusion between the E1 protein and the endosomal membrane. This event is followed by the disassembly of the viral nucleocapsid, releasing the positive-sense genomic RNA (gRNA) into the cytoplasm. The positive-strand genomic RNA serves as messenger RNA (mRNA) for the immediate translation of nonstructural proteins (nsP), generating the polyprotein precursor, P1234. The P1234 is an inactive precursor of the viral replicase complex. It is self-cleaved by the C-terminal cysteine protease region of nsP2, producing nsP4 that acts as RNA-dependent RNA polymerase (RdRp) and viral replicase complex consisting of individual non-structural proteins [36]. The nsP4 will synthesize the negative-strand RNA intermediate as a template for the synthesis of the positive-strand gRNA [37]. The viral replicase complex is also responsible for the production of the subgenomic viral RNAs (sgRNA) to direct the synthesis of the structural proteins (Figure 1) [38].

Four nsPs, along with the viral gRNA, and presumably host proteins, integrate at the plasma membrane (PM) to form viral replication compartments (spherules) containing viral double-strand RNA intermediate (viral dsRNA). In this spherule, nsP1-4 functions to generate gRNA, antigenomic (negative-strand RNA intermediate), and sgRNA. It is believed that the replication process in this spherule will protect the viral dsRNA from intracellular RNA sensors [39]. The sgRNA is then translated into a structural polyprotein precursor, C-p62-6K-E1. The capsid (C) protein has a serine protease domain capable of self-cleavage from the rest of the structural polyproteins, releasing the C protein to the cytoplasm. Thus, its cleavage does not depend on the host cell machinery [40,41]. Once produced, the C protein assembles with the gRNA to form the icosahedral nucleocapsid core. Multiple capsid binding sites have been identified in the gRNA of CHIKV to facilitate selective gRNA packaging. This event is the determinant step of the new viral production [42].

**Figure 2 viruses-15-00048-f002:**
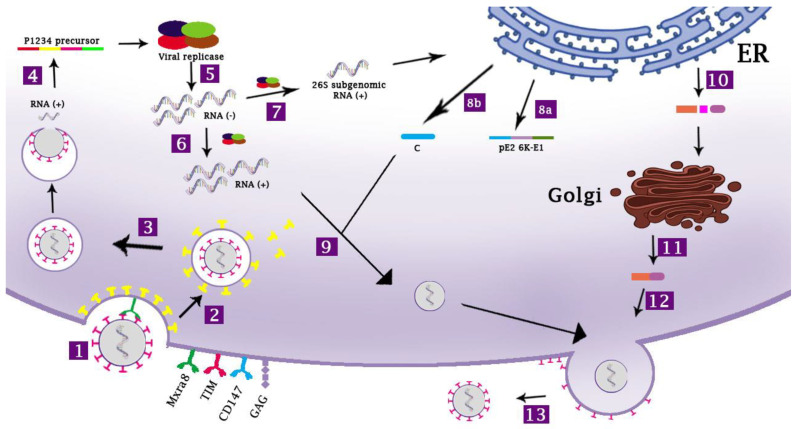
**The life cycle of CHIKV infection.** There are several potential receptors for CHIKV, including Mxra8, CD147, GAGs, and TIM **[1]**. CHIKV enters the target cells via clathrin-mediated endocytosis pathway **[2]**. Other entry pathways, such as micropinocytosis, are not depicted. Upon formation of the early endosome, clathrin molecules dissociate from the endocytic vesicle **[3]**. The pH acidification of endosome (endocytic vesicles) triggers the fusion of the endosomal membrane with the viral membranes (the E1 protein), releasing the genomic RNA, followed by an immediate translation of the non-structural polyproteins (P1234 precursor) by the ribosome **[4]**. The P1234 polyprotein is then cleaved by the nsP2, releasing the individual non-structural proteins, to form the viral replicase complex. The complex mediates the synthesis of the negative-strand RNA **[5]** that serves as templates for the synthesis of new positive-strand RNA **[6]** as well as for the synthesis of 26S subgenomic RNA **[7]**. The synthesis of negative-strand RNA intermediate, genomic, and subgenomic RNA occurs in the specialized replication compartments termed spherules (not depicted). The subgenomic RNA is subsequently translated into the structural polyprotein precursor C-pE2-6K-E1 in the rough endoplasmic reticulum (RER). The C protein contains a protease domain responsible for its self-cleavage. It dissociates from the polyprotein **[8b]** and assembles with the genomic RNA to form the icosahedral nucleocapsid core in the cytoplasm **[9]**. The pE2-6K-E1 precursor will be addressed to the lumen of the RER **[8a]** for maturation process **[10]**, culminating in the formation of E1-E2 heterodimers **[11]**. E1-E2 heterodimers will be inserted in the cell membrane forming the “virus budding microdomain” **[12]**. The assembled icosahedral nucleocapsid core migrates to this domain, and new viral particles will be extracellularly released by budding process **[13]**. See the main text for further details. The drawing is a modification of a figure that was previously published by Constant et al. [43].

The structural polyprotein processing and post-translational modification, including glycosylation, are conducted within the endoplasmic reticulum and Golgi apparatus [44]. The host proteases, such as furin protease, will cleave the p62-6K-E1 precursor to produce individual structural proteins, including E1, E2, E3, and 6K to build up the new viral particles [45]. The nucleocapsid core migrates to the plasma membrane regions containing the E1 and E2 proteins. The E1 and E2 assemble into trimers of heterodimers embedded in the viral membrane. Mature virions are released by the budding process from the infected cells [46]. Although not fully understood, this process is influenced by temperature, pH, and some of the host factors including cholesterol, actin, and tetherin [47].

## 3. The Viral and Host Factors Involved in the Pathogenesis of CHIKV Infection

Acute infection of CHIKV begins with viral transmission from the infected mosquito bite to the skin. After the skin bite, the CHIKV enters subcutaneous capillaries and replicates in susceptible cells, including macrophages, epithelial, and endothelial cells [48]. The virus subsequently spreads through the lymphatic system and bloodstream, leading to viremia, and reaches the sites of primary replication [39]. It has been demonstrated that human osteoblasts and human synovial fibroblasts were susceptible and permissive to CHIKV infection [49,50]. Following CHIKV-infected mosquito bites in humanized mice models, viral RNA was detected in bone marrow, liver, and lung, indicating viral dissemination in multiple organs [51]. In another mice model of CHIKV infection, a high inoculum dose resulted in viral persistence and chronic tissue damages [52]. It has been shown that in nonhuman primates (NHPs), macrophages were the main cellular reservoirs of CHIKV during the chronic phase of the disease [53]. Meanwhile in mice, CHIKV RNA was mainly detected in splenic B cells and follicular dendritic cells during the chronic phase of infection [54], suggesting their roles in maintaining viral persistence in the spleen.

During the acute phase of infection in children, the median viral copy number was 1.3 × 10^8^ copies/mL and could reach a maximum of 9.9 × 10^9^ copies/mL [55]. Meanwhile, in adult patients, the viral load was 3.85 × 10^6^ copies/mL, although some patients had a lower viral load (3.1 × 10^3^ copies/mL) [56]. The viral load could serve as a predicting factor of clinical severity. A higher viral load was associated with fever and arthralgia in adult patients [57]. Similarly in children, viral load was positively associated with fever [58]. Adult patients with higher viral loads tend to have more involvement of large and small joints and myalgia compared to those with lower viral load [57]. In contrast, pediatric patients with myalgia had lower viral load compared to those without myalgia [55]. Noteworthily, patients with high levels of viremia were more prevalent in hospitalized cases [59]. The levels of interleukin 6 (IL-6) and monocyte chemoattractant protein-1 (MCP-1) in the patient’s plasma could serve as reliable biomarkers of high viral load in patients with CHIKV [56].

Differences in CHIKV genotypes infecting the patients may influence the host immune responses and thus, could influence the disease pathogenesis. In one study, the Asian genotype had less replication capacity compared to the IOL genotype in in vitro infection employing primary mouse tail fibroblasts. Consistently, the viremia level was higher in the IOL genotype compared to the Asian genotype in experimental infection in mice [60]. The Asian genotype also induced weaker systemic proinflammatory responses, including interferon-α (IFN-α), interferon-γ (IFN-γ), tumor necrosis factor-α (TNF-α), and IL-6, as well as lower levels of natural killer (NK) cell activity compared to the IOL genotype, and these were associated with lesser joint pathology [60]. Another study assessed whether the Asian and the ECSA genotypes differed in neuroinvasiveness [61]. Intracerebral inoculation of both genotypes resulted in similar viral titers in the brains, although the Asian genotype had higher mortality rates compared to the ECSA. Remarkably, the gene expression profiles showed that the Asian genotype induced higher levels of proapoptotic genes, while the ECSA caused higher upregulation of antiapoptotic and antiviral genes as well as genes involved in the central nervous system protection [61]. Experimental infection of the WA, ECSA, IOL, and Asian strains in *IFNaR*^−/−^ mouse (A129) models demonstrated that different strains led to differential mortality rates in these mice [62]. In addition, the WA strain had higher viremia levels compared to the ECSA and IOL strains at one day post-infection [62]. In rhesus macaques, the IOL strain induced more robust antibody and T cell responses compared to the WA strain [63]. All of these findings suggest that various CHIKV genotypes may differ in virulence characteristics that may influence the disease progression and outcome.

The host genetic factors also contribute to the susceptibility and progression of viral infections and their clinical outcome. A previous study in rhesus macaque showed that aged animals mounted reduced B and T cell responses, leading to viral persistence [63]. In humans, it has been shown that polymorphism in *Toll-like receptors 7 (TLR7)* and *TLR8* genes were significantly associated with susceptibility to CHIKV infection [64,65]. *OAS*, *DC-SIGN (CD209)*, and *TLR3* gene polymorphisms were associated with the development of clinical symptoms in patients infected with CHIKV [66,67,68]. In addition, *TNFα* gene polymorphism was associated with chronic joint pain [67]. A recent systematic review revealed that *DRB1*14* was associated with the susceptibility of symptomatic CHIKV infection [69]. Since these genes are involved in viral recognition and subsequent antiviral IFN and adaptive immune responses, these associations highlight the importance of the host immune responses in human susceptibility to CHIKV infection and in modulating the disease pathogenesis, as described below.

## 4. The Antiviral IFN Responses against CHIKV Infection

The acute infection induces inflammatory responses and infiltration of monocytes, macrophages, neutrophils, and NK cells [70]. The infection of CHIKV induces systemic innate responses which primarily involved antiviral IFN-α, pro-inflammatory cytokines, and chemokines. The process is followed by the activation of the adaptive immune responses [71].

### 4.1. Stimulation of IFN Responses

Specific pattern recognition receptors (PRRs) for RNA viruses will recognize the presence of CHIKV infection within the target cells. RIG-I-like receptors (RLRs), including RIG-I and MDA5, sense the presence of viral RNA in the cytoplasm. In the endosomal compartment, viral RNA will be recognized by TLRs, including TLR3, TLR7, and TLR8. Following viral recognition by TLRs and RLRs, interferon regulatory factor 3 (IRF3), IRF7, and nuclear factor kappa-B (NF-κB) will be activated, leading to the induction of type I IFNs and various proinflammatory chemokines and cytokines [72].

TLR3 is involved in the antiviral IFN response against CHIKV [68]. Lack of TLR3 signaling in both human and mouse fibroblasts led to a notable increase in CHIKV replication in vitro. Similarly, *Tlr3*^−/−^ mice had a 100-fold higher viral load compared to wild-type mice and exhibited uncontrolled viral dissemination [68]. Consistently, the absence of TRIF, IRF3, and IRF7, among key downstream signaling pathways of TLRs, markedly increased CHIKV replication [68,73,74]. In plasmacytoid DCs, IRF7-mediated signaling was essential to protect mice against the lethal challenge of CHIKV via induced production of type I and II IFNs [75]. In addition to TLRs, RIG-I is also involved in antiviral responses against CHIKV. 5′pppRNA, a RIG-I agonist, was shown to inhibit CHIKV replication in human fibroblast MRC-5 cells [76]. Interestingly, this RIG-I-mediated protection was independent of the type I IFNs [76].

It has been shown that CHIKV infection triggers the induction of type I IFNs (IFN-α and IFN-β) and IFN-γ [74,77]. In several cell lines infected with CHIKV, including Vero, HFF-1, HT-1080, and SK-N-MC cell lines, IFN-α dose-dependently inhibited CHIKV replication [78,79]. However, although IFN-α and IFN-β share an identical receptor, they have distinct mechanisms for protecting against severe CHIKV diseases [73]. Early CHIKV replication and dissemination to multiple tissues during the acute stage were mainly inhibited by IFN-α, while IFN-β was predominantly involved in limiting neutrophil-induced inflammation [73]. IFN-α, but not IFN-β, was pivotal to controlling the progression of CHIKV into the chronic stage [80]. The primary roles of type I IFNs in protection against CHIKV-induced mortality were shown in mice deficient in IRF3 and IRF7 (IRF3/7^−/−^). CHIKV infection in this mouse model was lethal, and this was associated with the failure to induce IFN-α and IFN-β production [74]. Consistently, severe and lethal CHIKV infection was observed in mice lacking IFN-α/β receptors (IFNAR^−/−^) [74,81]. A closely related study showed that this type I IFN signaling acted via non-hematopoietic cells rather than on immune cells [82]. In another mouse model, nasal administration of adenovirus-vectored IFN-α both as a prophylactic and a therapeutic agent conferred effective protection against the lethal challenge of CHIKV infection [83]. Notably, it has been reported that excessive production of IFN-α was associated with poor clinical outcomes in patients infected with CHIKV [84]. While extensive studies have been performed to delineate the role of type I IFNs, type III IFNs (IFN-λ) were also shown to have antiviral effects against CHIKV replication [85]. Further studies are required to describe the precise role of type III IFNs during CHIKV infection, particularly in humans.

The increased production of type I IFNs will signal in an autocrine or paracrine manner to intensify the signal or to induce an antiviral state in neighboring uninfected cells, respectively. Type I IFNs will bind to IFN-α/β receptors and ultimately result in the induction of hundreds of IFN-stimulated genes (ISGs) via JAK-STAT signaling pathways. Subsequently, those ISGs cooperatively establish an antiviral state within the infected and adjacent cells, including inhibition of viral replication and virion maturation [72]. In human skin fibroblasts and in mice models infected with CHIKV, several ISGs were notably upregulated, including IFIT1, STAT1, IFIT3, and ISG15, among others [86,87], suggesting their functional roles in defending against CHIKV infection. Indeed, several studies have characterized the role of individual ISGs in restricting CHIKV replication. IFI16 was upregulated upon CHIKV infection, and its overexpression led to marked inhibition of CHIKV replication [88]. Other ISGs that have been demonstrated to exert anti-CHIKV effects were IFITM3 [89,90], DDX56 [91], IFIT1, IFIT2, IFIT3 [86], IRF1 [92], and ISG20 [93]. The essential roles of ISG expression were also shown in microbiome-depleted mice [94]. In these mice, microbiome depletion led to uncontrolled CHIKV dissemination due to altered TLR-MyD88 signaling that resulted in a striking reduction of type I IFN production by plasmacytoid DCs and ISG expression by monocytes [94].

### 4.2. Antagonism of IFN Responses by CHIKV

To survive within the infected host cells, CHIKV has evolved multiple strategies to evade the IFN responses, similar to many other viruses. CHIKV-encoded proteins, including nsP2, E1, and E2, strongly inhibited the activation of the IFN-β promoters by MDA5/RIG-I and MAVS signaling pathways [95]. Additionally, nsP2 suppressed IRF3-mediated activation of the IFN-β promoter [95] as well as MDA5/RIG-I and MAVS-mediated activation of NF-κB promoters [96]. In addition to nsP2, E1 and E2 proteins had similar inhibitory activities of the NF-κB promoter [96]. Thus, CHIKV-encoded proteins may evade viral recognition that leads to the dampening of IFN responses [97].

In Vero cells infected with CHIKV and treated with IFN-α, IFN-β, and IFN-γ at four hours post-infection, CHIKV replication was not inhibited [98]. This finding suggests that CHIKV was potentially resistant to IFN treatment in some cell lines in vitro. Further studies showed that this IFN resistance involved inhibition of the JAK-STAT signaling pathway. CHIKV-nsP2 was shown to inhibit the induction of ISGs. Mechanistically, nsP2 efficiently blocked the nuclear translocation of STAT1 [98,99]. nsP2 also promoted the nuclear export of STAT1 [100]. Interestingly, the presence of mosquito saliva also supported CHIKV replication by dampening the JAK-STAT signaling pathways that led to decreased expression levels of ISGs [101].

## 5. The Host Adaptive Immune Responses against CHIKV Infection

### 5.1. The T Cell-Mediated Immune Responses against CHIKV

An essential role of CD4 T cells in controlling CHIKV infection was shown in CD4-deficient mice. In these mice, the development of anti-CHIKV antibodies was suboptimal and exhibited reduced neutralizing capacities [102]. This finding indicates the requirement of CD4 T cells in mediating humoral immunity against CHIKV. The functionality of T cell responses was also studied in chronic and recovered CHIKV patients at 12 to 24 months post-infection. IFNγ-producing CD4 and CD8 T cells were detectable in the majority (85%) of the patients (Figure 3). This IFN-γ response was mainly directed against the nsP1 and E2 peptides, although the intensity was mainly induced by E2 [103]. More specifically, the C-terminal half of the E2 protein induced a high frequency of T cell response and the highest level of IFN-γ release [103].

However, the pathogenic roles of CD4 T cells were also shown. CHIKV infection in CD4^−/−^ and CD8^−/−^ mice resulted in similar viremia levels with the wild-type mice. Interestingly, joint swelling was significantly reduced in CD4^−/−^ mice. Characterization of the immune cells demonstrated that the numbers of CD4 T cells were significantly increased in the infected joints and partially controlled the infiltration of CD8 T cells, although the exact mechanism remained elusive [104]. This finding indicates that CD4, and not CD8 T cells, are responsible for the joint inflammation induced by CHIKV infection.

In patients with acute CHIKV, it was shown that CD8 T cells were activated, as demonstrated by increased expression of CD69 (an early activation marker), CD107a, perforin, granzyme, among others [105]. This pattern suggests that CD8 T cells are activated during the acute phase and functionally mediate cytotoxic activities. In the mice model, increased numbers of CHIKV-specific CD8+ T cells accumulated in the spleen and joint-associated tissues. These CHIKV-specific CD8 T cells were capable of producing IFN-γ upon ex vivo stimulation, indicating their functionality as effector T cells [106]. Interestingly, the presence of these functional effector T cells did not result in the reduction of viremia levels in CHIKV-infected mice. However, the presence of functional effector CD8 T cells prior to CHIKV infection resulted in CHIKV clearance, although this effect was more predominant in the spleen than in the joint-associated tissues [106]. Altogether, these data suggest that CHIKV has developed strategies to evade recognition of CD8 T cells to establish chronic infection in the joints.

It has been shown that manipulation of T cell responses could hold promise for future therapeutic intervention strategies. In a chronic CHIKV mouse model, administration of anti-CD137 resulted in viral clearance [54]. CD137 is a costimulatory molecule found on the surface of T cells, NK cells, and B cells. The therapeutic effect of anti-CD137 was partially CD4 and CD8 T cell-dependent and associated with the elimination of the CHIKV reservoir in B cells and follicular dendritic cells in secondary lymphoid tissues during the chronic phase [54].

Several studies investigated the function of regulatory T cells (Treg) in the CHIKV disease pathogenesis [107,108]. The frequency of Treg was lower in acute and chronic CHIKV compared to recovered cases. In line, the IL-10 level was higher in recovered than in acute and chronic cases [107]. In addition, impaired functions of Treg were reported, which were characterized by lower expression PD-1, CTLA-4, and TGF-β [108]. This finding suggests that during acute and chronic CHIKV disease, Treg frequency and function are impaired and that normalization of Treg is associated with disease resolution. In mice, Treg expansion led to the resolution of CHIKV-induced joint inflammation. Treg expansion induced energy of CHIKV-specific CD4 T cells and subsequent infiltration of CD4 effector T cells [109].

**Figure 3 viruses-15-00048-f003:**
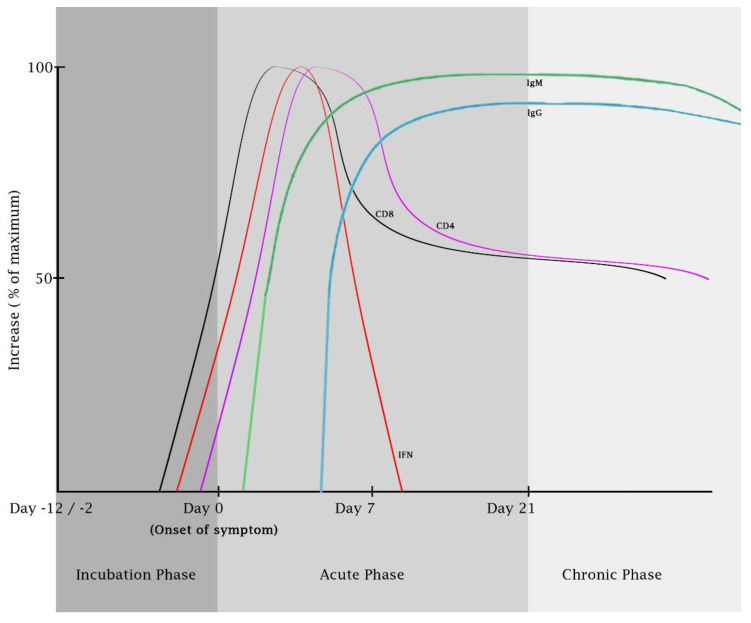
**The kinetics of immune responses against CHIKV infection.** The level of IFN peaks at between 0 and 5 after symptom onset [110]. The activation of CD8 and CD4 T cells peaks at day 1 and 4 after symptom onset, respectively. The functional CD4 and CD8 T cell responses are detectable at 12 to 24 months post-infection [71]. Anti-CHIKV IgM gradually increased from as early as day 2–4 after symptom onset and then remained stable until 4-10 months [111,112,113,114]. Anti-CHIKV IgG is detected after the first week of infection and remains positive for several years [115]. Since the drawing is based on limited number of studies, further comprehensive studies are required to delineate the precise kinetics and durability of immune responses against CHIKV infection.

### 5.2. The Humoral Immune Responses against CHIKV

An early study in mice showed the pivotal roles of antibodies to control CHIKV infection [102]. In B cell-deficient mice, CHIKV infection led to more severe diseases and viral persistence for more than one year. This finding indicates the essential role of antibody-producing B cells in mediating viral clearance [102]. Administration of monoclonal antibodies obtained from patients infected with CHIKV protected the mice against the lethal challenge of CHIKV [116]. Another study employed lipid-encapsulated mRNA encoding potent human-derived monoclonal antibodies (CHKV-24 mAb). Infusion of CHKV-24 successfully reduced viremia levels and protected the mice from clinical disease and lethality [117]. Similarly, the administration of monoclonal antibodies to CHIKV-infected rhesus macaques led to viral elimination and reduced joint pathology [118]. Remarkably, CHIKV antibodies generated against a particular genotype conferred protection against other genotypes [62,119]. All of these findings emphasized the importance of antibodies for the clearance of CHIKV infection.

In CHIKV infection, anti-CHIKV IgM gradually increased from as early as day 2–4 after symptom onset [112,113], and then remained stable until 4 months [114]. Anti-CHIKV IgG, dominated by the IgG3 subtype, can be detected in the early convalescent stage at 10 days post-symptom onset in a group of patients and remained detectable after 2–3 months post-symptom onset (Figure 3) [120]. Another group of patients developed anti-CHIKV later at 4–6 weeks post-symptom onset. The early IgG3 antibody formation (at day 10) was associated with reduced viremia levels and prevention of chronic and severe diseases [120]. Similarly, early production of IgM antibodies with neutralizing capacities was associated with reduced viremia levels [113]. The neutralizing activity of IgM was complementary to the early IgG antibodies but played an important role in days 4 to 10 post-symptom onset [113]. The presence of IgM and IgG was also associated with cytokine and chemokine levels, suggesting the role of antibodies in modulating the overall immune responses in CHIKV-infected patients [56,120].

The E2 glycoprotein is the main target of antibody responses, both in mice and humans [102,121,122,123]. In mice, epitopes located in the C terminus of the E2 glycoprotein are the main target of antibodies [102]. Characterization of antibody responses in patients infected with CHIKV found multiple monoclonal neutralizing antibodies targeting distinct epitopes in the E1 and E2 glycoproteins. However, only E2-specific antibodies protected the mice against the lethal challenge of CHIKV infection [116]. CHIKV-E2-specific antibodies can block viral fusion and release from the infected cells [123].

Several other studies investigated associations between the quality of antibody responses and disease pathogenesis. Longitudinal analyses of CHIKV-infected patients demonstrated that CHIKV-specific IgG binding avidity was increased over time, in line with enhanced neutralizing capacities [121]. A significantly higher antibody avidity against E1 and E2 proteins was found in acute patients compared to chronic patients [119]. This finding suggests that the quality of the CHIKV-neutralizing antibodies plays an important role in protection against chronicity, although it is not clear the required threshold for protection against chronic progression.

The possible roles of anti-CHIKV antibodies in immunopathology have been addressed by several studies. In vitro, CHIKV infection in the presence of anti-CHIKV antibodies resulted in enhanced attachment and viral replication. Consistently in mice, experimental CHIKV infection in the presence of sub-neutralizing concentrations of anti-CHIKV antibodies worsened the progression of the diseases [124]. These findings highlight that antibody-dependent enhancement (ADE) should be further investigated, particularly during the development of CHIKV vaccines. Other studies focused on molecular mimicry between CHIKV and human proteins. Two epitopes within the E1 protein had notable similarities with human protein. Administration of these peptides into mice led to inflammation in the muscle to a similar extent to CHIKV-injected mice [125]. Another study reported the development of Sjogren’s Syndrome (SS) associated with CHIKV infection. This was possibly due to molecular similarity between CHIKV proteome and SS autoantigens [126]. Further studies are highly required to dissect the role of CHIKV-specific antibodies in immune-mediated pathology.

## 6. The Clinical Manifestations of CHIKV Infection

### 6.1. Acute Phase of Infection

In the past centuries, dengue and chikungunya fever were often misdiagnosed despite the fact that the clinical symptoms are distinguishable on the basis of differences in the disease onset and sequelae following recovery from the acute infection [127]. CHIKV infection can be categorized into three stages: the acute stage (from day 1 to day 21), the post-acute stage (day 21 to 3 months), and the chronic stage (more than 3 months) [128]. During the acute stage, the infected patients may experience a viremic phase (5–10 days) and a post-viremic phase (6–21 days) [128,129].

The clinical manifestations of CHIKV infection have a wide variety, ranging from asymptomatic, mild symptoms, to severe disabling disease (Table 1) [129]. A study showed that the asymptomatic infection ranged from 3%-47% of all infected cases [130]. The median incubation period of CHIKV infection is 3 days, ranging from 2 to 12 days [131]. The acute viremic stage of CHIKV infection is characterized by abrupt onset of high-grade fever (often >39 °C), arthralgia, myalgia, headache, fatigue, nausea, vomiting, and arthritis [129,132,133,134]. Other symptoms, including conjunctivitis, exanthema, and edema may also occur. The exanthema can present as diffuse or focal skin rash [10,135,136]. This disease is self-limiting and, in most patients, will resolve in 7–10 days [129].

In the post-acute phase, symptoms are characterized by varied manifestations resulting from persisting initial acute symptoms [137]. During this phase, some symptoms including articular symptoms and fatigue persist, but fever will diminish [10]. Polyarthritis is usually symmetrical and involves small and large joints, including knees, ankles, hands, and wrist [138]. Moreover, periarticular involvement, including enthesitis, tenosynovitis, and bursitis can also occur [128]. Acute CHIKV infection may also trigger an exacerbation of the previous autoimmune arthritis history [139].

Severe symptoms, involving vital organs, may develop during CHIKV infection. Individuals with comorbidities, elderly populations, and infants have a higher risk to experience these severe symptoms [134,140,141,142]. These severe complications include encephalitis, encephalopathy, neuro-ocular disease (uveitis, retinitis, optic neuritis), myelopathy and myelitis, Guillain-Barré syndrome, myocarditis, hepatitis, acute interstitial nephritis, severe sepsis, septic shock, and multi-organ failure [141,143,144]. Perinatal CHIKV infection can cause sequelae such as microcephaly, cerebral palsy, and neurocognitive impairment [145,146].

The clinical manifestations of CHIKV infection showed different characteristics between children, adults, and older adults [134]. Arthritis, joint swelling, and joint stiffness are present more frequently in adults than in children [134]. On the other hand, arthralgia is common in children and adults [134]. Older adults tend to have an atypical or severe presentation due to their comorbidities, and the disease could induce acute decompensation of comorbid conditions [140]. In children, recurrent CHIKV infections, associated with fever and viremia, are commonly found [147].

### 6.2. Chronic or Persistent Phase of Infection

The chronic stage of CHIKV virus infection is characterized by persisting symptoms of more than three months after the initial diagnosis of acute infection. Chronic CHIKV disease is often limited to more distal joints due to persistent viral replication and inflammation. CHIKV persistence is found in some organs, including endothelial cells in the liver, mononuclear cells in the spleen, macrophages within the synovial fluid and surrounding tissues, and satellite cells within the muscles [39]. The mechanism of this persistence is not fully understood. Some persistent symptoms such as arthralgia, myalgia, and arthritis have been described and suggest the persistence of viral infection in the target organ or deleterious inflammatory mechanisms which cause tissue damage [70].

Some studies have been performed to examine the chronic symptoms of CHIKV virus infection. The joint symptoms usually resolve within 1–3 weeks. However, a study in Singapore showed 13% of the patients had chronic arthralgia 3 months after infection [148]. A study from La Réunion showed that arthritis was experienced by half of the patients 1 year after inclusion [149]. Another study showed arthritis appeared for 4 months in 33% of the patients, 20 months in 15%, and 3–5 years in 12% [150]. During this stage, patients could experience unpredictable relapses of fever, asthenia, arthralgia, and stiffness [150,151]. The older patients and those who had previous rheumatic or traumatic joint disorder were more susceptible to developing this chronic stage [150].

Researchers have studied the predictor factors for infected patients to develop chronic symptoms. A cohort study in the French West Indies (La Martinique) investigated the predictors for the occurrence of chronic CHIKV arthritis in 193 patients and found that age, female sex, and dehydration state during the acute phase as important factors [152]. Another study found that a high baseline viral load was detected in patients with chronic diseases [149]. However, this result was not in accordance with a study in Singapore [148]. The stronger inflammation in the acute phase is thought to be one predictor of the development of chronic symptoms. One study showed that a stronger inflammation, which was characterized by higher levels of C-reactive protein (CRP), was associated with higher levels of TNF-α, IL-8, IL-6, and IL-12, and did not significantly predict the development of chronic symptoms [149].

## 7. Diagnosis of CHIKV Infection

The diagnosis of CHIKV infection based on laboratory examination is very important due to its unspecific and overlapping signs and symptoms. The diagnosis of CHIKV merely based on clinical symptoms is highly challenging, particularly in endemic areas where other arboviruses, including DENV and Zika virus (ZIKV), are also co-circulating. CHIKV infection should be taken into consideration in patients presenting with acute febrile illness and polyarthralgia, especially travelers who recently returned from areas with known endemicity of CHIKV transmission [153]. A clinical study showed that fever and polyarthralgia had 84% sensitivity, 71% positive predictive value (PPV), and 83% negative predictive value (NPV) [154].

Laboratory examination of CHIKV infection can be performed using various methods, including viral isolation, serological methods, and RNA detection using real-time quantitative reverse transcriptase polymerase chain reaction (qRT-PCR) [155]. In the past, the diagnosis was mainly based on serological methods to detect the presence of specific antibodies against CHIKV. With the advance of molecular techniques, qRT-PCR is more commonly used to detect the presence of viral RNA, particularly in the acute phase of infection. Most importantly, the testing method should be carefully selected by considering the purpose of the examination and the timing of specimen collection [155]. Accordingly, the interpretation of laboratory findings should be based on the kinetics of viral replication (viremia) and antibody responses in humans (Figure 4).

### 7.1. Virus Isolation

Virus isolation by using cell culture is the gold standard for viral detection and it is highly specific [150]. The isolation of infectious viral particles is commonly used in research settings, such as for pathogenesis study and molecular characterization. The major limitation of viral culture is its low sensitivity, and the results require at least 48 h. Other limitations are that it requires an expensive cost, extended time, high-containment laboratory equipment (biosafety level 3), and skilled laboratory personnel [156,157]. Thus, virus culture is rarely performed in clinical (diagnostic) settings [157]. The isolation could be performed in the acute viremia stage, before the eight days of infection [157]. Noteworthily, the sensitivity could be increased when the viral culture is performed on the day or near the acute febrile onset where viremia reaches its highest level [158].

To isolate CHIKV, some cell lines from humans, monkeys, and mosquitos have been employed [159]. Several studies showed that Vero (African green monkey kidney epithelial cells), BHK-21 (baby hamster kidney fibroblast), C6/36 (*Aedes albopictus* cells), and BEAS-2B (bronchial epithelial) cell lines produced high titers of CHIKV, while low titers were found in RD (human rhabdomyosarcoma) and A549 (human alveolar basal epithelial) cells [48,160]. In addition, while human primary CD4 T lymphocytes and Cd14 monocytes do not support CHIKV replication, human primary macrophages productively support CHIKV replication [48].

CHIKV induces a marked cytopathic effect (CPE) during the culture which can be observed as early as 24 h post-infection [48]. A study that compared the diploid human embryonic lung (HEL) and Vero cell cultures to isolate CHIKV showed that CPE appeared earlier in the HEL than in Vero [161]. In HeLa cells, CHIKV generates extensive cell death by apoptosis which can be seen by using immunofluorescence assay and can be measured by using colorimetric MTT assay [48].

### 7.2. Nucleic Acid Detection

Nucleic acid detection is a rapid and highly sensitive assay to diagnose CHIKV infection. The qRT-PCR is a common method to detect the genome of CHIKV [162]. It can be used as multiplex PCR to simultaneously detect the presence of other arboviruses, including ZIKV [163]. Recently, loop-mediated isothermal amplification (LAMP) has been developed that enables rapid amplification of the viral genome under isothermal conditions without the need for expensive facilities such as thermocycler [164,165]. Thus, the major advantages of LAMP are it is simpler, cheaper (compared to qRT-PCR), sensitive, and rapid which enables its routine application in remote areas [164,165].

After primary CHIKV infection, the virus is replicating rapidly resulting in a high level of viremia. A systematic review estimated that the median incubation period is around 3 days (range 2–12 days) [131]. The viral RNA of CHIKV can be detected by the qRT-PCR method from 0 to 7 days of infection, after which qRT-PCR detection becomes unreliable [155,166]. A study examining the viremia profile of laboratory-confirmed CHIKV cases demonstrated that the virus RNA can be detected as early as 6 days prior and extend to 13 days post-acute fever onset. Viral RNA reaches its highest level at or near the onset of febrile illness (up to 6.1 × 10^8^ pfu/mL) [158]. In asymptomatic individuals, the viral load can be as high as 2.9 × 10^5^ pfu/mL, with a median of 3.4 × 10^3^ pfu/mL [166].

A conserved region of the envelope *E1* and *E2* genes is the most common target for qRT-PCR [2,167]. Other targets include *nsP1* and *nsP4* genes [168,169]. To diagnose CHIKV infection, plasma and serum have become the most commonly used clinical samples [170]. However, other body fluids, such as saliva, urine, vaginal secretion, and semen can also contain CHIKV during the acute phase of the disease [171].

### 7.3. Serology Tests

#### 7.3.1. IgM and IgG Antibodies-Based Serological Tests

Serology testing is a simpler testing than qRT-PCR and can be conducted to detect anti-CHIKV immunoglobulin M (IgM) and immunoglobulin G (IgG). These antibodies can be detected by enzyme-linked immunosorbent assay (ELISA), immunofluorescence assays (IFA), and plaque reduction neutralization tests (PRNT) [172]. The IgM ELISA is the most commonly used method to establish the diagnosis of CHIKV infection [173].

As mentioned, IgM can be detected in 2–10 days after the onset of infection and it remains detectable up to 4–10 months post-infection [111,112,113,114]. Meanwhile, IgG is detected after the first week of infection and remains positive for several years [115]. However, if the arthritis persists for >3 months after CHIKV infection, it is recommended to check rheumatoid factor (RF), anti-citrullinated peptide antibody (anti-CCP), and human leukocyte antigen (HLA) B27 to find possibilities underlying rheumatoid arthritis and spondyloarthritis [128].

There are various commercially available kits for IgM and IgG detections developed by Euroimmun (Lubeck, Germany), Standard Diagnostics (SD) Inc. (Yongin-si, Republic of Korea), Abcam (UK), and InBios (Seattle, WA, USA), either as ELISA-based, immunofluorescence assays (IFA)-based, and rapid tests. The overall sensitivity and specificity of ELISA- and IFA-based tests are high (more than 90%), although the sensitivity of SD Chikungunya IgM ELISA (Standard Diagnostics Inc., Yongin-si, Republic of Korea) is relatively low (65.3%) [174]. For rapid tests, the sensitivity is very low (27.9% for On-site CHIKM IgM Combo Rapid test (CTK Biotech Inc., San Diego, CA, USA) and 19.1% for SD BIOLINE Chikungunya IgM (Standard Diagnostics Inc., Yongin-si, Republic of Korea)) [174].

A systematic review and meta-analysis showed that IgM detection test had more than 90% diagnostic accuracy for the ELISA-based test, IFA, in-house developed tests, and samples collected after seven days of the symptom onset [174]. However, the sensitivity of IgM detection tests was lower for rapid tests (42.3%), commercial tests (78.6%), and samples collected seven days before the symptom onset (26.2%) [174]. Therefore, IgM detection tests are highly recommended for samples taken during the convalescent phase of CHIKV infection. The specificity of IgM detection tests was more than 90%, regardless of the test formats and time of sample collection [174]. Additionally, the diagnostic performance of the IgG test was more than 93% [174]. While many commercial ELISA kits are based on the whole-inactivated CHIKV, recombinant E2-based ELISA tests have also been developed [175,176].

Serology diagnosis is limited by cross-reactivities with other arboviruses [177]. CHIKV is antigenically similar to other viruses within the *Alphavirus* genus, including Semliki Forest Virus (SFV), MAYV, and ONNV [3,178]. In South America where CHIKV and MAYV cocirculate, the interpretation of serological tests was problematic due to high false positives [179]. Cross reactivities with DENV, which belongs to the *Flavivirus* genus, have also been reported, with an overall sensitivity of 100% and a very low specificity of only 25.3% [180]. Thus, reliable serology diagnostic tools should be more carefully evaluated in both acute and convalescent patient sera, particularly in regions where multiple viruses co-circulate [181].

#### 7.3.2. Antigen-Based Serological Tests

Serological tests for CHIKV E1/E2 antigen detection have also been developed, including rapid [112,182,183], ELISA-based test [184,185], and fluorescent-linked immunosorbent assay (FLISA)-based tests [186]. A meta-analysis study of rapid and ELISA-based tests showed that they have a good performance for clinical samples collected during the acute phase of infection [174]. The sensitivity and specificity of rapid-based tests were 85.8% and 96.1%, while the ELISA-based test was 82.2% and 96.0%, respectively [174]. Notably, a study reported that FLISA had higher sensitivity compared to ELISA [186].

The main challenge for developing antigen-based serological tests is the performance heterogeneity due to different CHIKV genotypes [187]. An initial evaluation of diagnostic accuracy of immunochromatographic-based rapid test with limited samples showed a higher sensitivity for the ECSA genotype (88.9%) compared to the Asian genotype (33.3%) [182]. However, the development of new monoclonal antibodies against the E1 protein has improved its sensitivity against the Asian genotype [185]. Thus, the identification of antibodies that specifically recognize conserved epitopes across CHIKV genotypes is necessary for further development of antigen-based test against CHIKV. These newly developed tests should be evaluated in different geographical settings to cover all circulating genotypes.

## 8. The Current Development of Vaccine Candidates against CHIKV

Since the first isolation of CHIKV, significant efforts have been invested to develop CHIKV vaccines. However, until recently, there are no CHIKV vaccines that have been approved to prevent the infection. Along with the progress of molecular techniques, new-generation platforms of CHIKV vaccine candidates have been developed with promising safety and efficacy profiles. Various vaccine platforms or delivery strategies can be employed to construct CHIKV vaccines, including classical approaches (inactivated, live-attenuated vaccine, and protein subunit vaccines) and novel approaches (recombinant virus-vectored vaccines, virus-like particles, and nucleic acid DNA or RNA vaccines) [188]. Importantly, some studies for the development of CHIKV vaccine candidates have entered phase II and/or phase III clinical trials (Table 2).

A previous study examined the potentiality of an E2 protein-based recombinant vaccine and a whole virus-inactivated vaccine [189]. Both vaccines induced an anti-CHIKV antibody response in a dose-dependent manner. Furthermore, both vaccines conferred protection to CHIKV-infected mice. Upon challenge, vaccinated mice had undetectable levels of viral load in blood and tissues [189]. E2_CHIKV_ recombinant protein formulated with an adjuvant Poly (I:C) induced efficient E2_CHIKV_-specific humoral and cellular immune responses [190]. However, the excellent safety profiles of inactivated and protein subunit vaccines come at the expense of their efficacy, since both vaccine platforms require good and immunogenic adjuvants [189,190,191].

**Table 2 viruses-15-00048-t002:** The summary of the recent CHIKV vaccine candidates in phase II/III clinical trials.

Vaccine	Platform	Background CHIKV Strain	Number of Doses	The Last Stage of Development	Developer	References
VLA1553	Live-attenuated	LR2006-OPY1 (ECSA-IOL)	Single dose	Phase III (completed)	Valneva, Austria GmbH	NCT04786444, NCT04838444, NCT04546724, NCT04650399
VRC-CHKVLP059-00-VP	Virus-like particle	37997 strain of the WA genotype	Two doses	Phase II	The US National Institute of Allergy and Infectious Disease (US NIAID)	NCT02562482, NCT01489358, and [192,193]
PXVX0317	Virus-like particle (adjuvanted)	37997 strain of the WA genotype	Single dose	Phase III	Emergent BioSolutions (Gaithersburg, MD, USA)	NCT03483961, NCT05072080, NCT05349617, and [194]
MV-CHIK-202	Recombinant measles virus-vectored vaccine	pTM-MVSchw-CE3E26KE1	Single or two doses	Phase II	Themis Bioscience GmbH	NCT02861586, NCT03101111, and [195,196]
BBV87	Whole virus-inactivated	ECSA	Two doses	Phase II and III	International Vaccine Institute	NCT04566484

### 8.1. Live-Attenuated Viral Vaccines (LAV)

LAV is highly immunogenic compared to the other platforms since the attenuated strain retains a weakened replicating capacity and thus, can induce stronger immune responses. The attenuated phenotype can be achieved by recombination technology and site-directed mutagenesis in both structural and non-structural proteins of CHIKV [197,198,199]. An attenuated phenotype of CHIKV was successfully achieved by site-directed mutagenesis of a nucleolar localization sequence (NoLS) in the N-terminal region of the capsid protein [198]. The resulting attenuated strain, termed CHIKV-NoLS, led to minimal inflammation and tissue damage in inoculated mice. The stability of the attenuated phenotype was confirmed by examining growth kinetics and plaque size following extended in vitro passage in Vero cells and long storage at −20 °C and −80 °C [200].

Another LAV candidate manipulated the nsP3 gene by deleting 60 amino acid residues in the P1234 polyprotein (∆5nsP3) in the background of the CHIKV LR2006-OPY1 strain of the ECSA-IOL genotype [199,201]. Preclinical data showed that this ∆5nsP3 was highly immunogenic and conferred protection against CHIKV disease in mice and cynomolgus macaques [199,201]. A phase I clinical trial showed that one dose of ∆5nsP3 (VLA1553) was well-tolerated and induced high and sustained seroconversion rates after a one-year follow-up [202]. Recently, VLA1553 (Valneva, Austria GmbH) has been reported to complete the phase 3 trial. Seroprotection of CHIKV-neutralizing antibodies was confirmed in 98.9% of participants after one month of receiving the single-shot dose [203]. In line, the passive transfer of sera from VLA1553-vaccinated volunteers to NHP conferred complete protection from CHIKV viremia following the challenge with wild-type (WT) CHIKV [204].

The previous production of LAV employs traditional culture methods in the cell culture systems [205]. There are several limitations of this “traditional” LAV, including variability of the characteristics of the vaccine products depending on the culture conditions used, contamination of vaccine materials, limited scalable production, and safety issue related to the continuous passages that lead to genetic changes (genetic drift). In addition, a safety concern of LAV formulation is the reversion to the pathogenic phenotype that may occur during the production process and post-administration in the vaccinees [205,206]. A considerable number of arthralgia in vaccinees was reported during a previous phase II trial of LAV CHIKV strain 181/25 (TSI-GSD-218), raising concern about this safety issue [205]. The attenuated phenotype of CHIKV strain 181/25 was largely attributed to a glycine-to-arginine 80 mutation at residue 82 (G82R mutation). To improve the safety of this vaccine candidate, muscle-specific miR-206 target sequences were incorporated into CHIKV strain 181/25 containing the G82R mutation [207]. Preclinical data demonstrated that the combined mutations led to enhanced safety in inoculated mice [207].

### 8.2. Nucleic Acid-Based (RNA) Vaccines

Recent advances in RNA vaccine technology have driven rapid development of many vaccine candidates targeting emerging viral diseases, such as Pfizer and Moderna vaccines which have been approved for the novel coronavirus disease 2019 (COVID-19) and they are now widely available for mass vaccination [208]. Additionally, it can be combined with another vaccine platform to improve its efficacy. For instance, recently developed candidates for CHIKV vaccines combine the advantages of contemporary RNA vaccine technology with the more established LAV. Administration of a full-length in vitro-transcribed live-attenuated CHIKV genome by employing nanostructured lipid carrier (NLC) via intramuscular injection in C57BL/6 mice led to induction of high titers of neutralizing antibodies and protection to the subsequent lethal challenge of CHIKV [209]. Similarly, a liposome RNA delivery system was employed to directly deliver the self-replicating RNA genome of CHIKV-NoLS into mice [210]

### 8.3. Virus-like Particle (VLPs)

Another CHIKV vaccine candidate employed virus-like particles (VLPs) as the main strategy to elicit protective immunity in humans. VLPs are generated by the expression of CHIKV structural genes by using a DNA expression plasmid transfected into human cells. The expressed proteins subsequently form viral particles that are similar to intact virions because of their self-assembly capacity. However, VLPs are replication-incompetent because of the absence of genomic RNA and thus, have a better safety profile compared to LAV [211].

A VLP vaccine, the VRC-CHKVLP059-00-VP (CHIKV VLP), has been developed by selective expression of viral structural proteins of the 37997 CHIKV strain of the WA genotype [212]. Preclinical data from nonhuman primate experiments showed that CHIKV VLP stimulated a high titer of neutralizing antibodies that conferred protection against the heterologous CHIKV challenge [212]. The vaccine was safe, well-tolerated, and elicited neutralizing antibody responses comparable with natural infection titers in a phase I trial in healthy individuals. It was also capable of neutralizing CHIKV strains of distinct genotypes [193,213]. A phase II clinical trial has been completed to assess the safety and tolerability of the CHIKV VLP vaccine candidate [192]. The data showed a good safety profile with no serious adverse events related to the investigational product. The vaccine candidate elicited a durable neutralizing antibody response during a follow-up of 72 weeks after vaccination [192]. Another phase II trial has been conducted to evaluate the immunogenicity and safety of PXVX0317, an aluminum hydroxide-adjuvanted formulation of the CHIKV VLP vaccine. It was shown that PXVX0317 induced a durable neutralizing antibody response against CHIKV after 2 years of follow-up [194].

### 8.4. Recombinant Virus-Vectored Vaccines

Recombinant virus-vectored vaccines are developed by manipulating the vector virus genome to contain the gene of interest encoding potential antigen derived from a specific virus [214]. The vector itself could be replication-incompetent (including modified vaccinia Ankara (MVA), influenza virus, and adenovirus (AdV)) or replication-competent viruses (including measles virus, influenza virus, vesicular stomatitis virus (VSV), and Newcastle disease virus (NDV)) [208]. One limitation of this approach is that its effectivity is influenced by the pre-existing host (human) immunity against the vector itself. However, this challenge can be circumvented by employing vectors that are rarely found in humans, vectors derived from animal viruses, or vectors that weakly induce immunity in humans [208].

A measles virus-vectored vaccine has been developed by inserting structural genes of CHIKV in the measles virus genome to express CHIKV VLPs [215]. In immunized mice, this vaccine elicited high titers of antibodies and protected the mice from the lethal CHIKV challenge [215]. This vaccine was further evaluated in cynomolgus macaques. High neutralizing antibody titers were induced in vaccinated animals that conferred protection against viremia and CHIKV-associated diseases [216]. Importantly, good safety and immunogenicity profiles were demonstrated during the phase I and II clinical trials and warrant its efficacy investigation in a phase III trial [195,196]. Other CHIKV vaccine candidates used AdV type 5 (AdV-5) and chimpanzee AdV (ChAdOx1) [217,218]. AdV-5-based vaccines expressing various combinations of CHIKV structural genes were administered intranasally in C57BL/6 mice and effectively stimulated neutralizing antibodies that protect the mice from viremia and CHIKV-associated pathology [218]. Similarly, ChAdOx1-based vaccines capable of expressing CHIKV VLP efficiently induced both T cell and antibody responses against CHIKV [217].

## 9. Conclusions and Future Perspectives

There is recently significant progress in scientific studies of various aspects of CHIKV. However, the remaining gaps need to be extensively addressed. For example, the true burden of CHIKV-associated diseases, particularly in developing countries, should be given more attention since there are overlapping clinical manifestations with other endemic viral infections in these regions. Thus, the development of standard diagnostic assays that are simple, inexpensive, and accurate would be important in these resource-limited settings. An improved understanding of the immunopathogenesis of chronic CHIKV disease is necessary to develop therapeutic interventions. In addition, there is no CHIKV vaccine available that has been used to prevent the infection. We believe that the availability of safe and effective CHIKV vaccines would significantly reduce the burden of chikungunya disease in the future.

## Figures and Tables

**Figure 1 viruses-15-00048-f001:**
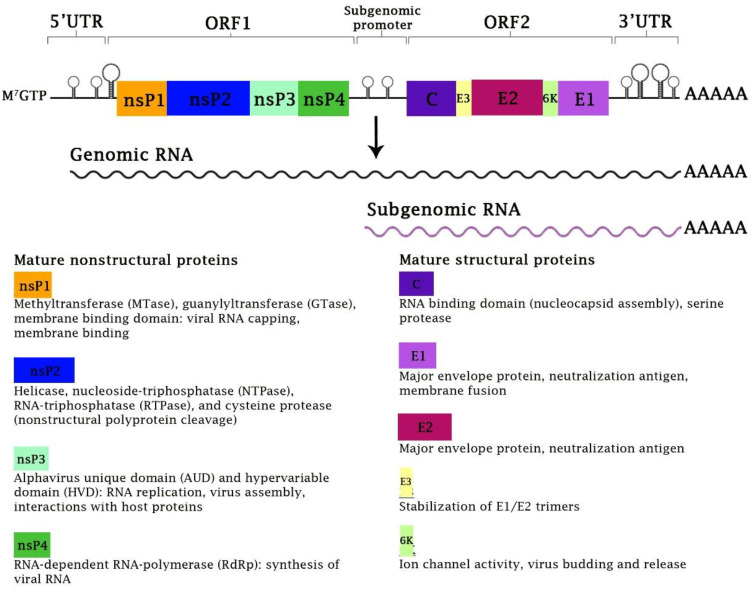
**The genome organization and viral proteins of CHIKV.** The CHIKV genome consists of a positive-sense single-stranded RNA molecule, approximately 11.8 kb in length with a 5′-methylguanylate cap (M^7^GTP) and a 3′-polyadenylate tail. The genome is divided into two ORFs, ORF1 and ORF2. ORF1 and ORF2 encode for the nonstructural and structural proteins, respectively. The expression of ORF1 is controlled by the genomic promoter located in the 5′-untranslated region (5′-UTR), while the expression of ORF2 is regulated by an internal subgenomic promoter. The genomic RNA is employed as template for synthesis of new viral genome for chikungunya virion assembly. The subgenomic RNA is used as a template for production (translation) of viral structural proteins. The known functions of individual CHIKV proteins are also depicted.

**Figure 4 viruses-15-00048-f004:**
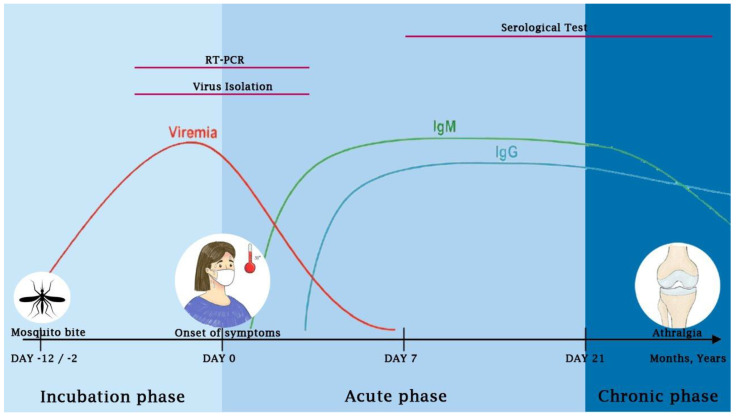
**The course of CHIKV infection and its association with different approaches to diagnose CHIKV infection.** The RT-PCR and virus isolation are best performed near the onset of febrile illness where viremia reaches its highest level. The serological tests to detect the presence of IgM and IgG are best performed after seven days of the symptom onset.

**Table 1 viruses-15-00048-t001:** The clinical manifestation of chikungunya fever.

Signs and Symptoms	Proportion	Reference
Fever	91%	[132]
Myalgia	64.9%	[133]
	61.0%	[132]
Arthralgia	100%	[133]
	86%	[132]
Arthralgia and myalgia	82%	[134]
Arthritis	58%	[134]
	56%	[132]
Back pain	55.0%	[132]
Rash	54%	[134]
	70.05%	[133]
Pruritus	61.9%	[133]
	12%	[132]
Conjunctivitis	4.2%	[136]
	21%	[132]
Non-severe hemorrhage	5%	[135]
Headache	47.1%	[135]
	74%	[134]
	69%	[133]
Fatigue	66%	[134]
Nausea	62%	[134]
	47%	[132]
Vomiting	60%	[134]
	21.0%	[132]
Diarrhea	12.0%	[132]
Retroorbital pain	18.0%	[132]
Abdominal pain	19.0%	[132]
Photophobia	9.0%	[132]

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
