# Peer review of "Understanding the Biology and Immune Pathogenesis of Chikungunya Virus Infection for Diagnostic and Vaccine Development"

_viruses, 2022, doi:10.3390/v15010048_

Round 1
Reviewer 1 Report
Hakim and Aman brings an updated revision of CHIKV infection with a comprehensive description of virus life cycle, IFN response against CHIKV and a general overview of chikungunya fever, diagnostic methods and vaccine development. Although there are some recent and excellent review papers in the literature regarding CHIKV infection, the present manuscript focus on a different aspect of the disease scenario, which is diagnostic methods and current vaccine candidates, which justifies its publication after some minor improvements.
1. As diagnostic and vaccine sections are those that distinguish this paper from others already published, I suggest the authors go a bit inside into details. I would appreciate to read a brief discussion about the commercially available kits, the differences in specificity and accuracy among them, what is the innovations in the theme... Regarding vaccines, it would be positive to include a table listing the major vaccine candidates in clinical trials together with the technology behind them and the major company/university leading the active.
2. Still regading CHIKV vaccine, there is already, at least, one candidate in Phase III trials. I suggest to include the public data about it (VLA1553). Otherwise, the review will not be as novel as it intend to be.
Minor revision: there are some typos, such as:
Page 2, line 44: "5' and 3'"
Page 7, line 262: "recognize"
Page 15, line 551: "equipment"
Page 17, line 620: "which"
Page 17, line 630: "specificity"
Author Response
Reviewer 1:
Q1. Hakim and Aman brings an updated revision of CHIKV infection with a comprehensive description of virus life cycle, IFN response against CHIKV and a general overview of chikungunya fever, diagnostic methods and vaccine development. Although there are some recent and excellent review papers in the literature regarding CHIKV infection, the present manuscript focus on a different aspect of the disease scenario, which is diagnostic methods and current vaccine candidates, which justifies its publication after some minor improvements.
Answer:
Thank you very much for your comments. In this revised version, we have improved our manuscript based on your suggestions and also suggestions of other reviewers.
Q2. As diagnostic and vaccine sections are those that distinguish this paper from others already published, I suggest the authors go a bit inside into details. I would appreciate to read a brief discussion about the commercially available kits, the differences in specificity and accuracy among them, what is the innovations in the theme... Regarding vaccines, it would be positive to include a table listing the major vaccine candidates in clinical trials together with the technology behind them and the major company/university leading the active.
Answer:
We have added a brief discussion about the commercially available kits and their performance (sensitivity and specificity) in page 19, line 634-642. I referred to a systematic review and meta-analysis by Andrew A et al. (PLoS Negl Trop Dis, 2022, PMID: 35120141). For development (innovations), we have mentioned them, for example the development of recombinant E2-based ELISA (page 19, line 651-653) and rapid test (page 18, line 664-666).
For vaccine development, we have updated the information in Table 2 (page 21) and revised the sentnce in page 20, line 691-692. We only list major vaccine candidates already in Phase II and/or III clinical trials.
Q3. Still regarding CHIKV vaccine, there is already, at least, one candidate in Phase III trials. I suggest to include the public data about it (VLA1553). Otherwise, the review will not be as novel as it intend to be.
Answer:
VLA1553 is a LAV (∆5nsP3), that we have discussed in the first version of the manuscript. Thank you very much for your information that this vaccine already completed phase 3 trial (in the first version, we mentioned that this vaccine completed phase 1 trial based on their publication in https://pubmed.ncbi.nlm.nih.gov/32497524/.) Thus, we have updated a discussion about VLA1553 in page 21-22, line 724-730.
Q4. Minor revision: there are some typos, such as:
Page 2, line 44: "5' and 3'"
Answer:
It has been revised (page 2, line 44).
Q5. Page 7, line 262: "recognize"
Answer:
It has been revised (page 8, line 274).
Q6. Page 15, line 551: "equipment"
Answer:
It has been revised (page 17, line 577).
Q7. Page 17, line 620: "which"
Answer:
It has been revised (page 19, line 658).
Q8. Page 17, line 630: "specificity"
Answer:
It has been revised (page 20, line 668).

Reviewer 2 Report
The review on chikungunya virus (CHIKV) by Hakim and Aman provides a clear, up-to-date and comprehensive review of the field. This is a valuable and easy-to-read review article that covers the biology, innate and adaptive immune responses, diagnostics and development of vaccines for CHIKV. As with any first submission of a large manuscript, there are a number of typographical errors and a few grammatical mistakes that need to be corrected (a careful re-read of the manuscript by the authors will likely catch most/all of these), but generally the authors have done a nice job. A few suggestions are noted below, the inclusion of which may prove beneficial.
1. A detailed figure of the genome and proteins produced would be valuable to the reader as well as a family tree to place CHIKV in context with other family members
2. Part 4 (details of interferon involvement) could be tightened. In a number of places in that section, there seems to be some redundancy and, overall, it was a bit unorganized. Some editing will improve the section.
3. Either expansion of current figure 2 to include the timing of all immune responses (including innate immunity and T cells – not just antibody responses) OR a new figure just focusing on the kinetics of the infection and immune response would be valuable. I understand that this is not the goal of the current figure, but inclusion of this additional information into the current figure 2 might be possible. Otherwise, inclusion of another figure would provide this graphical information.
4. Discussion of the different genotypes and the potential impact of these genetic changes on CHIKV diagnostics is limited to a single sentence. It would be nice to expand this discussion a bit.
Further, the authors mention the issue of serological cross reactivity of Ab responses to other related viruses, but there is no mention of the important cross reactivity with Mayaro virus in South America that makes serological tests problematic in this geographic region. A section expanding on these ideas would benefit the review.
Author Response
Reviewer 2:
Q1. The review on chikungunya virus (CHIKV) by Hakim and Aman provides a clear, up-to-date and comprehensive review of the field. This is a valuable and easy-to-read review article that covers the biology, innate and adaptive immune responses, diagnostics and development of vaccines for CHIKV. As with any first submission of a large manuscript, there are a number of typographical errors and a few grammatical mistakes that need to be corrected (a careful re-read of the manuscript by the authors will likely catch most/all of these), but generally the authors have done a nice job. A few suggestions are noted below, the inclusion of which may prove beneficial.
Answer:
Thank you very much for your valuable comments. We have comprehensively checked the whole manuscript. Typographical errors were already corrected, as also shown by reviewer 1 and 3.
Q2. A detailed figure of the genome and proteins produced would be valuable to the reader as well as a family tree to place CHIKV in context with other family members
Answer:
In the revised version, we have included Figure 1 (page 3) to describe the genome and proteins produced by CHIKV and their known functions. However, for family tree, we decided not to include it since in this revised version we already have four figures and two tables (in contrast to the first submission that we have only two figures and one table). In addition, the figure of family (phylogenetic) tree is not within the scope of our article since it is mainly used to describe evolutionary relationship between virus species and indeed, we do not discuss them in our article.
Q3. Part 4 (details of interferon involvement) could be tightened. In a number of places in that section, there seems to be some redundancy and, overall, it was a bit unorganized. Some editing will improve the section.
Answer:
We have checked and removed some redundancies (sentences) in this section to improve it (line 267-330). Overall, we organize this section as following: (1) the first paragraph describes the general overview of PRRs for sensing RNA viruses; (2) the second paragraph describes the TLR involved in CHIKV recognition and its associated downstream signaling pathways that lead to IFN production; (3) after that, the third paragraph describes the role of different types of IFNs in antiviral responses against CHIKV; and (4) the fourth paragraph describes the role of ISGs because its expression is stimulated by IFN.
Q4. Either expansion of current figure 2 to include the timing of all immune responses (including innate immunity and T cells – not just antibody responses) OR a new figure just focusing on the kinetics of the infection and immune response would be valuable. I understand that this is not the goal of the current figure, but inclusion of this additional information into the current figure 2 might be possible. Otherwise, inclusion of another figure would provide this graphical information.
Answer:
Thank you very much for your valuable comments and suggestions to include one figure about timing of all immune responses. We do agree with the reviewer the importance of the figures to improve our understanding on the kinetics of the immune responses against CHIKV. We add figure 3 (page 12) to include the kinetics of major immune effectors described in our article (IFN, CD4 and CD8 T cells, and IgM/IgG responses). Unfortunately, to our knowledge, there are limited studies about the durability of immune responses other than antibody responses. Therefore, in the figure legends we mentioned: “Since the drawing is based on limited number of studies, further comprehensive studies are required to delineate the kinetics and durability of immune responses against CHIKV infection.”
Q5. Discussion of the different genotypes and the potential impact of these genetic changes on CHIKV diagnostics is limited to a single sentence. It would be nice to expand this discussion a bit.
Answer:
We expand the discussion in a new paragraph (page 20, line 673-681).
Q6. Further, the authors mention the issue of serological cross reactivity of Ab responses to other related viruses, but there is no mention of the important cross reactivity with Mayaro virus in South America that makes serological tests problematic in this geographic region. A section expanding on these ideas would benefit the review.
Answer:
In the revised version, we have mentioned this issue (page 19, line 656-662).

Reviewer 3 Report
This review describes the current knowledge of chikungunya virus infection. This is very well written review and will provide useful summarization of current status of the field. More detailed description of molecular biology of CHIKV will greatly improve the quality of this review. Additional figure showing viral genome structure, function of viral proteins, and response to host defense will be needed to understand the biology of CHIKV and clinical management of CHIKV infection. Genotypes of CHIKV need more comprehensive attention. The current knowledge of the effects of difference of CHIKV genotypes in detection, virulence, immune response, and vaccine development are the necessary content.
Minor points:
In line 46; Use AUL for Asian clade for consistency.
In line 245; polymorphims > polymorphism
In line 249; polymorphims > polymorphism
In line 330-331; these data are somewhat contradictory with the data stated in line 294-296. Please add discussion.
In line 254-255; please clarify the interaction between CD4 T cells and CD8 T cells
In line 402-403; the timeline is different with Figure 2.
In line 715; which 3 genotypes? There are 4 main genotypes.
Author Response
Reviewer 3:
Q1. This review describes the current knowledge of chikungunya virus infection. This is very well written review and will provide useful summarization of current status of the field. More detailed description of molecular biology of CHIKV will greatly improve the quality of this review. Additional figure showing viral genome structure, function of viral proteins, and response to host defense will be needed to understand the biology of CHIKV and clinical management of CHIKV infection. Genotypes of CHIKV need more comprehensive attention. The current knowledge of the effects of difference of CHIKV genotypes in detection, virulence, immune response, and vaccine development are the necessary content.
Answer:
Thank you very much for your comments. To have a more detailed description of molecular biology of CHIKV, we added Figure 1 (page 3) to show the viral genome structure, as well as the known functions of each structural and nonstructural protein.
We include figure 3 (page 12) to include the kinetics of major immune effectors described in our article (IFN, CD4 and CD8 T cells, and IgM/IgG responses), as also suggested by another reviewer. Unfortunately, to our knowledge, there are limited studies about the durability of immune responses other than antibody responses. Therefore, in the figure legends we mentioned: “Since the drawing is based on limited number of studies, further comprehensive studies are required to delineate the kinetics and durability of immune responses against CHIKV infection.”
We also added the influence of different CHIKV genotypes in the antigen detection (page 20, line 673-681), as also suggested by another reviewer. We have discussed the influence of different genotype on immune response (immune pathogenesis) (page 7, line 232-253) and vaccine development (page 20, line 767-781) in the first version of the manuscript.
Q2. In line 46; Use AUL for Asian clade for consistency.
Answer:
We have added in page 2, line 46.
Q3. In line 245; polymorphims > polymorphism
Answer:
It has been revised (page 8, line 257).
Q4. In line 249; polymorphims > polymorphism
Answer:
It has been revised (page 8, line 261).
Q5. In line 330-331; these data are somewhat contradictory with the data stated in line 294-296. Please add discussion.
Answer:
The data stated in line 294-296 were derived from in vivo experimental models (in mice), while the data in line 330-331 were derived from in vitro in one specific cell lines (Vero cells). We have modified the interpretation by revising the sentence in page 10, line 341-342: (“This finding suggests that CHIKV was potentially resistant to IFN treatment in some cell lines in vitro.”)
Q6. In line 354-355; please clarify the interaction between CD4 T cells and CD8 T cells
Answer:
This finding come from observation that In vivo depletion of CD4+ partially reduced CD8+ T cell numbers as demonstrated by Teo et al. (J Immunology, 2013, PMID: 23209328). However, the exact mechanism is not clear, and therefore, we modified the sentence by adding: “although the exact mechanism remained elusive”. (page 11, line 365-366).
Q7. In line 402-403; the timeline is different with Figure 2.
Answer:
Thank you very much for this comment. We have revised the sentence and cite the associated references investigating the appearance and persistence of IgM/IgG antibodies during CHIKV infection (page 13, line 428-429).
Q8. In line 715; which 3 genotypes? There are 4 main genotypes.
Answer:
We have revised the sentence to avoid misinterpretations: “It was also capable of neutralizing CHIKV strains of distinct genotypes.” (page 23, line 773-774).
